# HARNESSING TEXT-TO-IMAGE DIFFUSION FOR DENSE PREDICTION TASKS

## ABSTRACT

Equipped with large-scale training data, text-to-image diffusion models have demonstrated the capacity to generate high-quality images that semantically correspond to the given textual descriptions. These compelling results imply that visual semantic knowledge has been effectively encapsulated within the generative diffusion model. The prospect of utilizing this embedded knowledge as a prior for down-stream vision tasks presents an intriguing avenue for exploration, which remains notably under-investigated. In this work, we demonstrate that when provided with appropriate image tags as textual descriptions, the implicit knowledge within these text-to-image diffusion models can be effectively leveraged for visual dense prediction tasks. Initially, we discover that supplying ground-truth semantic labels as textual instructions significantly enhances performance due to the extracted high-quality visual knowledge. Motivated by this observation, when presented with noisy tagging labels, we propose an adapter module attempting to derive relevant semantic information. Subsequently, we propose a multi-label classification learning objective which further enriches the semantic quality of tags, thereby amplifying the efficacy of knowledge extraction. We conduct extensive experiments four benchmarks, which suggest that the proposed approach is effective to unlock the representational capabilities of text-to-image diffusion models, showcasing a promising avenue for advancing dense prediction tasks in visual domains.

## 1 INTRODUCTION

In the current wave of advancing generative models, the domain of Natural Language Processing (NLP) has experienced notable progress, illustrated by models such as GPT (Radford et al., 2018; 2019; Brown et al., 2020), T5 (Raffel et al., 2020), and PaLM (Chowdhery et al., 2022), which have exhibited outstanding performance across a variety of tasks. In contrast, the realm of computer vision is still navigating through its foundation models, and has not yet attained a similar level of success. However, leveraging large-scale pre-trained datasets, text-to-image generative models (Saharia et al., 2022; Rombach et al., 2022) have recently demonstrated remarkable capability in generating high-quality images that semantically correspond to the given textual descriptions. This indicates that diffusion models have acquired a level of visual understanding of images from high-level image granularity to low-level pixel granularity. Dense visual prediction tasks, such as semantic segmentation and panoramic segmentation, also requires high-level visual understanding of images of regions in order to obtain accurate classification of pixels. It is intriguing to explore the methodologies of extracting the latent embedded knowledge encapsulated within the diffusion model for these visual dense prediction tasks, which is still notably under-investigated.

Recent studies have revealed that text-to-image diffusion models, when pretrained with textual inputs as conditions, are capable of developing distinct representation features that align with the specified prompts and instruction (Hertz et al., 2022; Parmar et al., 2023). Following research (Baranchuk et al., 2021; Xu et al., 2023; Zhao et al., 2023) has built upon these models, employing diffusion models as the foundation model and adapting them to different visual tasks. However, a pivotal question remains: how can the embedded knowledge be effectively extracted for visual tasks, particularly for visual dense prediction tasks?

Following previous studies, we delve into examining the influence of textual inputs on the performance of dense prediction tasks when using text-to-image diffusion models as a the foundation model. Intuitively, we hypothesized that the accuracy of text would directly correlate with the quality of extracted knowledge, and subsequently, the performance in down-stream visual tasks. To test this, we conducted an "oracle" experiment where ground-truth semantic lass labels were employed as conditions to adapt a stable diffusion model (Rombach et al., 2022) for downstream tasks. The results, depicted in Figure 1, highlight the pivotal role of semantic condition for the efficacy of extracting knowledge from text-to-image models, thereby enhancing performance on subsequent downstream tasks. In comparison to an "unconditioned" setting, using accurate semantic class labels resulted in a substantial $+20$ mIoU improvement on ADE20K. In the case of other datasets, the text-to-image model also achieved state-of-the-art performance.

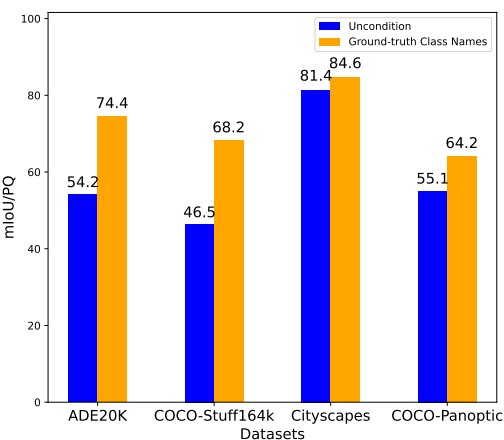

Figure 1: The comparison between the 'uncondition' and 'ground-truth class names' conditions, which reveals the impressive potential of text-to-image diffusion models in dense prediction when provided with the correct tags.

A significant performance disparity is observed between models operating without conditions and those conditioned on ground-truth semantics. Given the typical unavailability of accurate tags in real-world applications, it becomes intriguing to approximate the ground-truth semantic condition, with the aim of enhancing performance in downstream tasks. Specifically, we delve into and experiment with two strategies for approximating ground-truth semantics:

1) Utilize off-the-shelf zero-shot tagging models to identify or assign image tags. Specifically, we resort to pre-trained image tagging models to predict tags in a zero-shot setting. Even when the tagging space of the pre-trained data does not align with the semantic label space of downstream datasets, textual embeddings generated by pre-trained language models generally encapsulate semantic information (Raffel et al., 2020), which can be directly leveraged.

2) Incorporate a multi-label classification learning objective to further enrich the semantic quality of tags. Essentially, we train the tagging adapter to predict image tags. We employ this strategy in an effort to reduce the noise level in the zero-shot tagging model, and thereby approximate the ground-truth semantic condition more closely. Subsequently, these predicted semantic tags are fed into the diffusion model as conditions, which are hypothesized to be closer to the ground-truth semantic condition.

The two strategies we proposed significantly enhance the performance of diffusion models in dense predictions. Importantly, they can be used together, further boosting performance. Exhaustive experiments across various benchmarks, including semantic segmentation datasets like ADE20K (Zhou et al., 2019), COCO-Stuff164k (Caesar et al., 2018), and Cityscapes (Cordts et al., 2016), as well as the panoptic segmentation standard COCO-Panoptic (Lin et al., 2014), demonstrate that our approach consistently surpasses alternative text-to-image diffusion model transfer methods.

## 2 RELATED WORK

### 2.1 TEXT-TO-IMAGE GENERATION

Text-to-image generation endeavors to create convincing images inspired by textual descriptions. Reed et al. (Reed et al., 2016) laid the groundwork in this area by introducing the Conditional GAN. Subsequent advancements have achieved superior image quality via techniques including attention mechanisms (Xu et al., 2018), contrastive methods (Zhou et al., 2022; Zhang et al., 2021a), and multi-stage generation architectures (Zhang et al., 2017). One of the noteworthy strides in this field is the integration of diffusion models such as Stable Diffusion (Rombach et al., 2022), which

innovatively combine diffusion processes within the generative model framework. These models often utilize denoising autoencoders to approximate the inverse dynamics of a Markovian diffusion process (Sohl-Dickstein et al., 2015; Ho et al., 2020). A key characteristic of Stable Diffusion is its proficiency in generating visual content that aligns closely with textual descriptions, leveraging transformer architectures trained on vast datasets like LAION-5B (Schuhmann et al., 2022).

## 2.2 GENERATIVE REPRESENTATION LEARNING

Generative models have been widely used for crafting discriminative representations, especially within the realm of Generative Adversarial Networks (GANs) (Goodfellow et al., 2020). For instance, Big-BiGAN (Donahue & Simonyan, 2019) showcased impressive results on ImageNet recognition tasks (Deng et al., 2009). Concurrently, models like DatasetGAN (Li et al., 2022a; Zhang et al., 2021c) have illustrated the potential of GANs in enhancing visual perception tasks.

The recent trend emphasizes the power of diffusion models for discriminative representation learning. Initiatives like DDPM-Seg (Baranchuk et al., 2021) have combined unconditional diffusion denoising features with decoders to excel in segmentation tasks. Likewise, ODISE (Xu et al., 2023) leveraged a static diffusion model as a foundation for mask generation, establishing a benchmark in open-vocabulary panoptic segmentation. Remarkably, this model has seamlessly incorporated an implicit captioner, converting image features into cross-attention strata, thereby surpassing methods dependent on unconditional inputs. Meanwhile, VPD (Zhao et al., 2023) recommended initiating with a visual perception foundation anchored in pre-trained weightings and subsequently fine-tuning the denoising UNet using specialized decoders.

Inspired by these pioneering efforts, we believe that the vast potential of pre-trained text-to-image diffusion models remains untapped, largely due to the limited exploration of the pivotal of textual semantics. Consequently, our research aims to elucidate the influence of textual semantics, maintaining a rigorous yet clear methodology suitable for academic discourse.

## 3 METHOD

### 3.1 DIFFUSION MODEL OVERVIEW

This section provides a concise review of the latent diffusion model adopted in our study. We utilize the pre-trained latent diffusion model presented in (Rombach et al., 2022), which has undergone training through diffusion processing on vast text-image paired datasets. In its standard form, these models integrate a noise sample into a latent variable $z$ to produce $z_t$, formulated as:

$$z_t = \sqrt{\bar{\alpha}_t} x + \sqrt{1 - \bar{\alpha}_t} \epsilon \tag{1}$$

where $\alpha_1...\alpha_t$ are noise schedule hyperparameters, with $\alpha_t = \prod_{k=1}^{t} \alpha_k$. The training objective can be expressed as:

$$L_{LDM} := \mathbb{E}_{\varepsilon(x),c,\epsilon \sim \mathcal{N}(0,1),t} \left[ \|\epsilon - \epsilon_\theta(z_t, t, T(c))\|_2^2 \right] \tag{2}$$

where $T(c)$ signifies encoded text prompts, and $\epsilon_\theta$ commonly adopts a U-Net architecture, which will be optimized during the training process.

### 3.2 DIFFUSION FEATURES EXTRACTOR

The generative process of diffusion models is essentially the inverse of training, beginning with a noise distribution sampled from a Gaussian distribution (Song et al., 2020; Ho et al., 2020; Karras et al., 2022). Although diffusion models are well-known for producing high-resolution images using multi-step denoising mechanisms, they are not specifically designed for dense prediction tasks. For instance, dense prediction commonly starts with a specific image rather than Gaussian noise. To adapt diffusion models for such tasks: 1) Use a VQGAN encoder (Esser et al., 2021) to extract latent image features. 2) Introduce minor noise to these features, which, in combination with textual prompts, feeds into a pre-trained denoising U-Net. 3) Capturing the U-Net's internal features, denoted as $f_i(\epsilon_\theta, z_t, T(c))$. 4) Feed the acquired features into a task-specific decoder and compute the discrepancy between predicted outcomes and the actual ground truth:

$$L = D(f_i(\epsilon_\theta, z_t, T(c))) \tag{3}$$

During training, one can choose to either freeze the original diffusion model parameters or fine-tune them. Empirical results suggest that the latter approach usually yields enhanced performance.

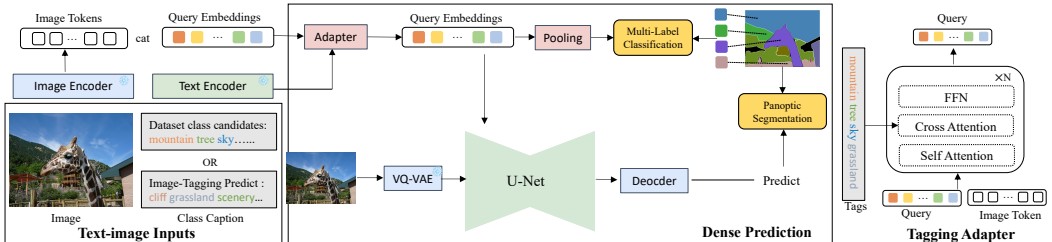

Figure 2: The overall framework for our method. (a) Given an image, we first formulate image-text pair inputs. The text can be derived from one of two methods: using the full class candidates related to the datasets or employing off-the-shelf image tagging models to predict image tags. These pairs are then fed into the frozen image encoder and text encoder. (b) A set of queries is introduced to the tagging adapter with $\times N$ attention blocks. This process can be supervised using a multi-label classification loss against the ground-truth labels. Subsequently, these queries are treated as diffusion conditions, guiding the diffusion model to procure features relevant to downstream tasks.

## 3.3 CONDITION ADAPTER

For the diffusion model, conditioning plays a pivotal role in determining semantic content within internal features. In the generative pre-training phase, $\epsilon_\theta$ is optimized with respect to the joint distribution of $(z_t, t, T(c))$. Here, $z_t$ is a noisy rendition of $z = \text{VQGAN}(x)$. Identifying the ideal textual condition $T(c)$ for dense prediction tasks remains an area of active research. Potential strategies include:

1) Unconditional input: Using an empty text prompt. Though not optimal, it's more favorable than resorting to irrelevant captions.

2) Off-the-shelf image caption models: Such as BLIP (Li et al., 2022b), which often overlook essential object details, leading to mediocre outcomes.

3) Training adapters for downstream tasks: Notably, the text adapter (Zhao et al., 2023) and the image-to-implicit caption adapter (Xu et al., 2023) are prevalent. The text adapter processes dataset-associated category names through a static text encoder, refined further by MLP layers:

$$T(c) = \text{TextEnc}(c) + \gamma \text{MLP}(\text{TextEnc}(c)). \tag{4}$$

On the other hand, the image-to-implicit caption adapter generates implicit captions from static image features:

$$T(c) = \text{MLP}(\text{ImageEnc}(I)). \tag{5}$$

## 3.4 TAGGING ADAPTER

While both text and image adapters present distinct advantages, neither fully harnesses the capabilities of pre-trained weights. This limitation primarily stems from their inability to supply the diffusion model with sharp, precise information. As highlighted in Figure 1, having accurate information can markedly boost the performance of the diffusion model across diverse datasets. However, obtaining ground-truth class labels during inference remains a challenge. To address this, we introduce a tagging adapter to extract tag information.

A straightforward approach involves using off-the-shelf image tagging models. Image tagging becomes particularly useful when ground-truth labels are inaccessible. This process predicts multiple labels for an image, often providing more detailed class information than other captioning models. Interestingly, our findings indicate that alignment between the label space of the image tagging model and the datasets is not mandatory. This adaptability allows for the integration of pre-trained tagging models with diverse label spaces, paving the way for zero-shot predictions on specific datasets.

However, these zero-shot predictions often produce tags that can be noisy. Directly employing such noisy labels without further refinement might result in a performance drop when compared to an approach without textual conditions. To mitigate this, we propose a tagging adapter enhanced with cross-modal attention, as visualized in Figure 2. This enhanced adapter employs learnable queries to facilitate attention mechanisms across both image and text features before they are integrated into the diffusion U-Net. This can be mathematically represented by:

$$c = \{c_i \in \text{Tag}(I)\}$$
$$T(c) = \text{MLP}(Q, \text{TextEnc}(c), \text{ImageEnc}(I)) \tag{6}$$

where $Q$ denotes the query embeddings and $\text{Tag}(I)$ signifies the predicted tags associated with the given image $I$.

Additionally, when ground-truth category labels are accessible during training, we can integrate a multi-label classification learning objective. We start by extracting query embeddings using Equation 6. Following an average pooling applied to the resultant query embeddings, the consolidated features are directed to a multi-label classifier. The weights of the classifier are initialized from the class embeddings and remain unchanged. The predicted labels can be computed as:

$$y_k = \frac{e^{(\text{Pool}(T(c)), h_k)}}{e^{(\text{Pool}(T(c)), h_k)} + 1} \tag{7}$$

where $y_k$ stands for the $k$-thlabel from the entire candidate set, and $h_k$ represents the classifier's $k$-th label weight. We adopt the asymmetric loss (Ridnik et al., 2021b) to fine-tune the tagging adapter, aligning with established practices. This loss function is perceived as conventional since the contrasting predicted query embeddings intrinsically highlight relevant specifics of the correct image classes.

## 4 EXPERIMENTS

This section provides a comprehensive description of our experimentation, detailing the implementation process, a comparative analysis with state-of-the-art methodologies for both semantic and panoptic segmentation, and an ablation study to highlight the significance of the proposed approach.

### 4.1 IMPLEMENTATION DETAILS

**Architecture:** Our core architecture utilizes the Stable-Diffusion v1.5 as the backbone. Throughout the experimental evaluations, the encoder from VQGAN (Esser et al., 2021) remains frozen while the U-Net (Ronneberger et al., 2015) is fine-tuned. We extract multi-scale features from the U-Net's up-sampling stages, consistent with the configurations outlined in (Zhao et al., 2023). These features exhibit dimensions of $[1280, 1280, 640, 320]$ and are shaped as $[8 \times 8, 16 \times 16, 32 \times 32, 64 \times 64]$. For the image and text encoders in our adapter, we employ a frozen CLIP-L/14 (Radford et al., 2021). To maintain architectural simplicity, we utilize either SemanticFPN (Kirillov et al., 2019) or UperNet (Xiao et al., 2018) as the default decoder for semantic segmentation tasks, as will be explicitly specified in our results section. For panoptic segmentation tasks, Mask2Former (Cheng et al., 2022) serves as our decoder, with $N = 100$ mask predictions. By default, we use RAM (Zhang et al., 2023) as our off-and-shelf zero-shot image tagging models.

**HyperParameters:** For the ADE20k (Zhou et al., 2019) dataset, we conduct experiments under two distinct settings: SemanticFPN for 80K iterations and UperNet for 160K iterations. The learning rates are set to $6 \times 10^{-5}$ for both 80K iterations and 160K iterations. The default tagging adapter's number of queries is 32, and what is a block, it has never been mentioned before: the number of blocks is 2. For panoptic segmentation tasks, the default learning rate is $1 \times 10^{-4}$. The batch size is 64 and trained with 9k iterations. The multi-label classification loss weight in both experimental settings is set to one.

## 5 COMPARISON WITH STATE OF THE ARTS

**ADE20k Benchmark** The ADE20k benchmark is celebrated for its comprehensive understanding of scenes, capturing a rich array of semantic details from 150 unique object and stuff categories. The

| Method | Pre-train Data | Crop Size | SemanticFPN mIoU | SemanticFPN +MS | UperNet mIoU | UperNet +MS |
|---|---|---|---|---|---|---|
| **Supervised pre-training** | | | | | | |
| PVTv2-B2 (Wang et al. 2022) | IN-1K | $512 \times 512$ | 45.2 | 45.7 | - | - |
| Swin-B (Liu et al. 2021) | IN-1K | $512 \times 512$ | 46.0 | - | 48.1 | 49.7 |
| Twins-SVT-L (Chu et al. 2021) | IN-1K | $512 \times 512$ | 46.7 | - | 48.8 | 49.7 |
| ViT-B (Dosovitskiy et al. 2020) | IN-1K | $512 \times 512$ | 46.4 | 47.6 | 46.1 | 47.1 |
| ConvNeXt-B (Liu et al. 2022) | IN-22K | $512 \times 512$ | - | - | 49.9 | - |
| InternImage-B (Wang et al. 2023) | IN-1K | $512 \times 512$ | - | - | 50.8 | 51.3 |
| Swin-L (Liu et al. 2021) | IN-22K | $640 \times 640$ | - | - | 52.1 | 53.2 |
| RepLKNet-31L (Ding et al. 2022) | IN-22K | $640 \times 640$ | - | - | 52.4 | 52.7 |
| ConvNeXt-XL (Liu et al. 2022) | IN-22K | $640 \times 640$ | - | - | 54.0 | - |
| InternImage-XL (Wang et al. 2023) | IN-22K | $640 \times 640$ | - | - | 55.0 | 55.3 |
| **Masked Image Modeling pre-training** | | | | | | |
| MAE-ViT-L/16 (He et al. 2022b) | - | - | 53.6 | - | - | - |
| BEiT-B (Bao et al.) | MM | $640 \times 640$ | - | - | 53.1 | 53.8 |
| BEiT-L (Bao et al.) | MM | $640 \times 640$ | - | - | 56.7 | 57.0 |
| **Multi-Modal pre-training** | | | | | | |
| CLIP-ViT-B (Radford et al. 2021) | MM | $640 \times 640$ | 50.6 | 51.3 | - | - |
| ViT-Adapter-Swin-L (Chen et al. 2022) | MM | $512 \times 512$ | 54.2 | 54.7 | 55.0 | 55.4 |
| **Diffusion pre-training** | | | | | | |
| VPD (Zhao et al. 2023) | LAION-2B | $512 \times 512$ | 53.7 | 54.6 | - | - |
| Ours | LAION-2B | $512 \times 512$ | 55.8 | 56.9 | 55.3 | - |
| Ours | LAION-2B | $640 \times 640$ | 56.2 | 57.2 | 56.8 | 57.4 |

Table 1: ADE20K val benchmark. 'IN-1K/22K' means ImageNet-1K/22K. MM means multi-modal pre-training. LAION-2B means the large-scale multi-modal dataset. '+MS' means multi-scale testing. SemanticFPN and UperNet are the different segmentation decoders. SemanticFPN is trained for 80K iterations, and UperNet is trained for 160k iterations

dataset consists of 20k training images complemented by a 2k-image validation set. We adopted the mean intersection over union (mIoU) as our performance metric. A detailed comparison with leading models is presented in Table 1, highlighting various models distinguished by their backbones and training datasets. Default, we use both zero-shot prediction labels and multi-label classification loss.

**Supervised pre-training** A dominant strategy for dense prediction tasks is supervised pre-training, including models such as InternImage-XL (Wang et al., 2023), tailored specifically for computer vision. Our method, when paired with the UperNet, achieves an increase of approximately +1.8 mIoU for single-scale testing and +2.0 mIoU for multi-scale testing. While supervised pre-training approaches exhibits robustness, they are frequently constrained by the availability of pre-trained data, given the high costs associated with acquiring supervised annotations. Our results indicate that, with right tagging adapter, large-scale pre-trained text-to-image diffusion models can potentially rival their supervised counterparts.

**Masked Image pre-training and Multi-Modal pre-training** Our model was benchmarked against the MAE-ViT-L/16 (He et al., 2022b) and CLIP-ViT (Radford et al., 2021). Our method consistently outperforms the baselines. Notably, we also drew comparisons with the BEiT-L (Bao et al.) model, which is a leading competitor that first uses self-supervised multi-modal data, then fine-tuning on the ImageNet-22K (Ridnik et al., 2021a) data. Within the UpperNet setting, our approach surpassed the BEiT-L model as well.

**Diffusion Pre-Training** The VPD is constructed upon the stable diffusion model v1.5. Notably, it incorporates the entire set of candidate class names when feeding input to the adapter. Using a similar SemanticFPN decoder configuration, our model achieved an increase of +2.1 mIoU under single-scale feature testing setting and an increase of +2.3 mIoU for multi-scale feature testing setting. This results highlight the importance of condition information for extracting knowledge from diffusion model.

| Method | Backbone | mIoU | +MS |
|---|---|---|---|
| OCRNet (Yuan et al. 2020) | HRNet-W48 (Sun et al. 2019) | 40.4 | 41.7 |
| OCRNet (Yuan et al. 2020) | HRFormer-B (Yuan et al. (2021)) | - | 43.3 |
| SegFormer (Xie et al. 2021) | MiT-B5 (Xie et al. 2021) | - | 46.7 |
| SegNeXt (Guo et al. 2022) | MSCAN-L (Guo et al. 2022) | 46.5 | 47.2 |
| RankSeg (He et al. 2022a) | ViT-L | 46.7 | 47.9 |
| UperNet-RR (Cui et al. 2022) | Swin-B | 48.2 | 49.2 |
| Segmenter (Strudel et al. 2021) | ViT-L | 49.1 | 50.1 |
| UperNet (Xiao et al. 2018) | BEiT-L | 49.7 | 49.9 |
| VPD* (Zhao et al. 2023) | SD (Rombach et al. 2022) | 48.3 | - |
| Ours | SD | 50.6 | 51.6 |

Table 2: COCO-stuff164k val benckmark. Ours method are trained with crop size of $640 \times 640$ and with 80k interations. * means our implement.

| Method | Backbone | Decoder | Crop Size | mIoU |
|---|---|---|---|---|
| Segformer | MiT-B5 | Mask2Former | 1024*1024 | 82.4 |
| Panoptic-DeepLab | SWideRNet | Mask2Former | 1024*2048 | 82.2 |
| Mask2Former-T | Swin-T | Mask2Former | 512*1024 | 81.7 |
| Mask2Former-L | Swin-L | Mask2Former | 512*1024 | 83.6 |
| OneFormer | DiNAT-L | Mask2Former | 512*1024 | 83.1 |
| VPD* | SD | SemanticFPN | 512*1024 | 81.8 |
| Ours | SD | SemanticFPN | 512*1024 | 82.6 |

Table 3: Cityscapes val benckmark. Our method is trained with 90k iteration2 with a lightly SemanticFPN decoder.

**COCO-Stuff164k Benchmark** The COCO-Stuff164k benchmark is a challenging dataset, comprising 171 unique classes, divided into 80 "thing" categories and 91 "stuff" categories.

As shown in Table 2, our approach consistently outperforms many top-tier models, such as SegFormer (Xie et al., 2021), RankSeg (He et al., 2022a), and Segmenter (Strudel et al., 2021). Notably, RankSeg utilizes a jointly-optimized multi-label classifier. The efficacy of RankSeg is closely tethered to the recall of its predictions, as omitted labels can result in a reduced decision space, potentially compromising performance. Unlike RankSeg, our model adeptly leverages predicted labels within cross-attention mechanisms, which can help mitigate the effects of inaccurately predicted labels. These experimental results confirms the effectiveness and robustness of our model in segmentation tasks.

**Cityscapes Benchmark** Cityscapes focuses on intricate urban scenes and encompasses 19 unique categories. Table 3 presents a comparative analysis of our approach against other leading models in this field. Our model outperforms VPD, which is also based on the Stable Diffusion model. The results again suggest the effectiveness of the proposed model. Our model slightly lags behind Mask2Former-L, given that the latter employs a more advanced decoder compared to the SemanticFPN we use. Meanwhile, the Cityscapes dataset's class variety is narrow, doesn't fully utilize our tagging adapter's potential (even in our oracle experiment in Figure 1).

**COCO-Panoptic Benchmark** The COCO-Panoptic dataset is a challenging collection containing 133 classes. We compare with baselines using metrics such as panoptic quality (PQ), mIoU, and mean average precision (mAP) in Table 4 By default, we employ the Mask2Former decoder for this benchmark. Our proposed model exhibits competitive performance across the board, surpassing several established methods in this task. This indicates the robustness and effectiveness of the techniques and strategies incorporated into our model.

When using the SD backbone, our method outperforms ODISE, especially in terms of PQ and mIoU. In the 100-queries setting, our method outperforms competitive models like Mask2Former and Panoptic SegFormer.

## 6 ABALTION STUDY

To verify the effectiveness of our model design, in this section, we examine the influence of the multi-classification learning objective, the zero-shot image tagging model (RAM), the number of adapter blocks, and the weights of the classification loss. All these ablation studies are conducted on the ADE20k dataset with a fixed input resolution of $512 \times 512$.

### 6.1 THE COMPARISON OF DIFFERENT ADAPTERS

Table 5 shows the performance of different adapters. We started with 'uncondition' input, which encodes empty semantic conditions to the diffusion U-Net. So, 53.9 could be seen the baseline performance. When solely conditioned on whole set of class labels, models like VPD offer an competitive performance. Yet, ODISE further enhances the performance by implicit captioner with $CLIP_{img}$. Furthermore, it's intriguing to note that the performance of Tag2Text-caption is worse than the baseline model. This discrepancy might be attributed to presence of noisy or incorrect semantic condition associated with the zero-shot captioning in Tag2Text. Such noises can potentially hinder the model's ability to accurately segment the images, underscoring the importance of reliable tagging in the zero-shot scenario.

Our proposed approach, which amalgamates both $CLIP_{img}$ and $CLIP_{text}$, consistently outperforms other strategies. This highlights the complementary of image and text-based cues in semantic segmentation tasks. The integration of a multi-label learning objective in our model leads to a tangible boost in performance (from 54.8 to 55.5), signifying the efficacy of such a loss in capturing the intricate nuances of the ADE20K dataset.

The addition of the RAM (zero-shot image tagging model) further augments our model's capabilities, culminating in an mIoU of 55.8 the highest among the models under consideration.

### 6.2 THE INFLUENCE OF LOSS WEIGHTS AND NUMBER OF BLOCKS

Table 6 and Table 7 presents a comprehensive analysis of our model's performance under varied configurations, focusing on the influence of different blocks and the weight of the loss function. As evidenced by the left part of Table 7, varying the weightage of the loss function has a distinct impact on the model's mIoU score. Interestingly, a weight of 5 yields the optimal mIoU of 55.72, which is marginally superior to other weight configurations. This suggests that a delicate balance is required when determining the loss weight, as both under-weighting and over-weighting can detrimentally affect the model's segmentation capabilities.

Turning our attention to the right section of Table 7, it's evident that the number of blocks plays a pivotal role in the model's performance. With 2 blocks, our model achieves an mIoU of 55.51, which stands as the highest among the considered configurations.

However, as we increase the number of blocks, a slight decline in performance is observed. This may imply that beyond a certain point, the addition of more blocks might introduce complexity without a

| Method | Backbone | PQ | AP | mIoU |
|---|---|---|---|---|
| DETR Carion et al. (2020) | R50 | 43.4 | - | - |
| K-Net Zhang et al. (2021b) | R50 | 47.1 | - | - |
| Panoptic SegFormer Li et al. (2022c) | PVTv2-B5 | 54.1 | - | - |
| MaskFormer Cheng et al. (2021) | Swin-B | 51.1 | 37.8 | 62.6 |
| Mask2Former Cheng et al. (2022) | Swin-T | 53.2 | 43.3 | 63.2 |
| Mask2Former | Swin-B | 55.1 | 45.2 | 65.1 |
| Mask2Former(200 quries) | Swin-L | 57.8 | 48.6 | 67.4 |
| FocalNet-L (200 quries) Yang et al. (2022) | Swin-L | 57.9 | 48.4 | 67.3 |
| ODISE Xu et al. (2023) | SD | 55.4 | 46.0 | 65.2 |
| Ours | SD | 56.1 | 46.5 | 66.5 |

Table 4: COCO-Panoptic val benchmark. Our method is trained with batch size 64 and 9k iterations, which is the same with ODISE.

| Method | Extra Captioner | Multi-Label Loss | mIoU |
|---|---|---|---|
| Uncondition | $CLIP_{text}$ | - | 53.9 |
| Tag2Text-Caption | Tag2Text (Huang et al. 2023) | - | 53.5 |
| VPD* | $CLIP_{text}$ | - | 54.2 |
| BLIP | BLIP (Li et al. 2022b) | - | 54.2 |
| ODISE* | $CLIP_{img}$ | - | 54.5 |
| Ours | $CLIP_{img}$ +$CLIP_{text}$ | - | 54.8 |
| Ours | $CLIP_{img}$ +$CLIP_{text}$ | yes | 55.5 |
| Ours | $CLIP_{img}$ +$CLIP_{text}$+RAM | yes | **55.8** |

Table 5: the influnce of different adapter on ADE20K, the setting is for 80K interations. * means our implement

| Adapter | Loss weight | mIoU |
|---|---|---|
| | 0 | 54.84 |
| | 1 | 55.51 |
| TextEnc +ImageEnc | 5 | **55.72** |
| | 10 | 55.04 |
| | 15 | 54.96 |

Table 6: The influence of different multi-label classification loss weight on ADE20K; the setting is for 80K iterations; TE means $CLIP_{text}$ and IE means $CLIP_{img}$.

| Adapter | Blocks | mIoU |
|---|---|---|
| | 2 | **55.51** |
| | 4 | 55.20 |
| TextEnc +ImageEnc | 6 | 55.43 |
| | 8 | 54.94 |
| | 10 | 54.73 |

Table 7: The influence of different adapter blocks on ADE20K; Adapter Block is showed in Figure.2 the setting is for 80K iterations. TextEnc means $CLIP_{text}$ and ImageEnc means $CLIP_{img}$.

corresponding increase in representational power, potentially leading to overfitting or diminished generalization.

While our model exhibits commendable performance across varied configurations, it's essential to juxtapose these results against those of other state-of-the-art models. The consistent outperformance of our approach reiterates the robustness and versatility of our model, especially when benchmarked against models that employ different conditioning strategies or loss weightages.

## 7 LIMITATION

Though text-to-image diffusion models demonstrate impressive capabilities in synthesizing high-quality images from textual descriptions, and hold potential for dense prediction tasks, there are inherent limitations. One primary constraint is their reliance on precise class tagging information.

The accuracy of the downstream tasks are deeply tied to the clarity and correctness of textual descriptions or image class tags. Ambiguities, inaccuracies, or contextual gaps in these descriptions can substantially undermine the model's performance. Furthermore, the model's adaptability across a spectrum of intricate real-world scenarios is yet to be validated, leading to questions about its robustness and adaptability.

## 8 CONCLUSION

This paper delves into the potential capability of text-to-image diffusion models for dense prediction tasks. By leveraging large-scale pre-training data, these models have showcased their ability to produce high-quality images based on varied textual descriptions. Our research indicates that with the right semantic conditions, the implicit knowledge within these models can be successfully applied to subsequent visual perception tasks. Experimental results reveal the significant role of ground-truth semantic conditions. Inspired by this observation, we propose a tagging adapter. This adapter is designed to offer robust and accurate semantic conditions, further enhanced by a multi-label classification loss function. Comprehensive evaluations across various benchmarks highlight the efficacy of the tagging adapter, demonstrating that the diffusion model can achieve superior results in visual dense prediction tasks.

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
