# OpenReview forum: "Harnessing Text to Image Diffusion for Dense Prediction Tasks"
_ICLR.cc/2024/Conference — ICLR 2024 Conference Withdrawn Submission_

### Official Review · Reviewer_6BRj · 2023-10-30

**Soundness:** 2 fair
**Presentation:** 1 poor
**Contribution:** 2 fair
**Rating:** 3
**Confidence:** 4

**Summary:**

This work highlights the efficacy of text-to-image diffusion models and believes that the embedded visual semantic knowledge (inside these models) can benefit dense prediction tasks (i.e., transfer knowledge from pre-trained generative models to discrimination models). The study presents a method to do this knowledge transfer for visual dense prediction tasks, utilizing appropriate image tags and an adapter module to improve performance, and thus, advancing dense prediction tasks.

**Strengths:**

This paper presents a clear motivation that an accurate semantic condition is important to extract knowledge from text-to-image models. The authors introduce a method to enhance the semantic condition, aiming to improve the performance of dense predictions with the pre-trained diffusion models.

**Weaknesses:**

1. Needing clarifications.
a) In the oracle experiment in the introduction, it's unclear which model was employed. Could the authors specify this?
b) In Section 3.4, the authors state that “our findings indicate that alignment between the label space of the image tagging model and the datasets is not mandatory.” What is the evidence supporting this claim? I understand that the adapter and multi-label classification tasks collaboratively facilitate this alignment. Could the authors elaborate on this?
c) In Equation (7), the term (Pool(T(c)),h_k) is ambiguous. I assume it represents a similarity function, but this needs clarification.

2. Ablation study. The paper lacks an ablation study related to query embedding. Such a study would provide insights into the significance and impact of this component.

3. Efficacy of RAM. One highlighted contribution is incorporating an off-the-shelf zero-shot tagging model, RAM. However, its effectiveness seems questionable. Table 5 indicates a mere 0.3% improvement, raising concerns about the computational overhead introduced by RAM versus its benefits.

4. Marginal improvements. Table 5 indicates a total performance improvement of 1.8%. While 0.9% of this gain is attributed to the introduction of CLIP_img (as proposed by ODISE), the proposed multi-label loss and RAM account for only 0.7% and 0.3% improvements, respectively. When compared with the substantial 21.7% enhancement observed on COCO-Stuff164k by using ground-truth class names (as shown in Figure 1), it appears that the authors' efforts to enhance the semantic condition might not be effective.

5. Unclear formulations. For Eq.(1), notations are not defined, which is not friendly to readers who do not know the latent diffusion process.

6. Poor writing quality. Author discussion comments are not removed from the draft (e.g., below Table 1, there is "yiqi: default, we use both zero-shot prediction ....") and there is an author name before the comments (against the anonymous policy of ICLR'24?). The overall writing is not easy to follow, and one guess is that ChatGPT was used to smooth the writing with long and complex sentences but with confused logic.

**Questions:**

See weaknesses.

**Details Of Ethics Concerns:**

Author discussion comments are not removed from the draft (e.g., below Table 1, there is "yiqi: default, we use both zero-shot prediction ....") and there is an author name "yiqi" before the comments. This might be against the anonymous policy of ICLR'24.

---

### Official Review · Reviewer_Mq6h · 2023-10-31

**Soundness:** 2 fair
**Presentation:** 2 fair
**Contribution:** 2 fair
**Rating:** 3
**Confidence:** 4

**Summary:**

The paper proposes to adapt text-to-image diffusion models for downstream dense prediction tasks. To this end, the paper proposes to use image tags as textual descriptions. To deal with noisy tagging labels, an adapter module to derive relevant semantic information is proposed along with a multi-label classification learning objective. Evaluation is performed on ADE20K,  COCO-stuff164k, and CityScapes.

**Strengths:**

·         The proposed method seems to show somewhat improved performance over VPD (Zhao et al. 2023) in ADE20K and COCO-stuff164k benchmarks.

·         The paper includes in Table 5 detailed ablations highlighting the performance improvement enabled by each component. Table 6 and 7 also ablates loss weights and adapter block sizes.

·         Figure 2 provides a good overview of the proposed approach.

**Weaknesses:**

The proposed method shows limited novelty over VPD (Zhao et al. 2023). The main contribution seems to be the addition of zero-shot tagging models to improve text embeddings. However, this addition of zero-shot tagging models along with a multi-label loss shows limited improvement over the use of off-the-shelf CLIP encoders in Table 5. Furthermore, the difference to text encodings obtained from the text adapter of Zhao et al. (2023) and the image-to-implicit caption adapter of Xu et al. (2023) should be explained in more detail.

·         The comparison to VPD (Zhao et al. 2023) is limited. Additional experiments, e.g., image segmentation on RefCOCO and depth estimation on NYUv2 as in Zhao et al. 2023 would be helpful.

·         The architecture of the tagging adapter is not well motivated. The TextEnc and ImageEnc in the the tagging adapter as described in Eq. 6 should be explained in more detail, including details of it model architecture. How does the TextEnc and ImageEnc relate to the cross attention modules in Figure 2.

·         The paper should include qualitative examples highlighting examples where the proposed approach outperforms VPD (Zhao et al. 2023).

·         Additionally, the paper is not well written:

o   In Figure 1, it is not clear what task is being performed, the model employed, and, the training and evaluation protocol.

o   The definition of T(c) is not clear. In Sec 3.1 it is described as “signifying encoded text prompts” and in Sec 3.3 it is described as referring to “dataset associated category names”.

o   In Sec 3.3 it is not clear what “dataset associated category names” refers to. Furthermore, the paper should distinguish between the terms “tag information”, “dataset associated category names” and “labels” used in sections 3.3 and 3.4.

o   In Section 3.4 it is not clear what “sharp, precise information” means in this context.



·         Typos: “lass” (page 2, para 1), “uncondition” (page 2, Figure 1), “k-thlabel” (page 5, para 2). Additionally, the use of \citep{} and \cite{} is not consistent.

**Questions:**

·         The paper should discuss in more detail the difference to prior work, e.g., VPD (Zhao et al. 2023).

·         The paper should provide more details of the text adapter, including details of the TextEnc and ImageEnc.

·         The paper should use a consistent citation style.

---

### Official Review · Reviewer_x6HC · 2023-10-31

**Soundness:** 3 good
**Presentation:** 3 good
**Contribution:** 2 fair
**Rating:** 5
**Confidence:** 5

**Summary:**

The authors show that text-to-image diffusion models can be effectively leveraged for visual dense prediction tasks when provided with appropriate image tags as textual descriptions. They first observe that supplying ground-truth semantic labels as textual instructions significantly enhances performance. Motivated by this observation, they propose an adapter module to derive relevant semantic information from noisy tagging labels. They also propose a multi-label classification learning objective to further enrich the semantic quality of tags.

**Strengths:**

- Exploring the conditional adapters in diffusion models for dense prediction is a new topic.

**Weaknesses:**

- This paper claims that using a text-to-image model for dense prediction but the main modification is on the adapter for the labels. So is it necessary to use a diffusion model for segmentation? Are these adapters useful on other baselines?
- Using diffusion models for dense prediction is not a very new topic. Please compare the proposed model with some recent works, like DDP [1].

[1]Ji Y, Chen Z, Xie E, et al. Ddp: Diffusion model for dense visual prediction[J]. arXiv preprint arXiv:2303.17559, 2023.

**Questions:**

Please refer to the weakness.

---

### Official Review · Reviewer_Sgbx · 2023-11-01

**Soundness:** 2 fair
**Presentation:** 2 fair
**Contribution:** 3 good
**Rating:** 5
**Confidence:** 4

**Summary:**

The paper leverages text-to-image diffusion models to extract dense image features and demonstrates the importance of the text prompts. The paper to generate the textual prompts by using an (zero-shot) image tagging model and propose an attention module to further improve the text prompts. The paper validates the effectiveness on semantic and panoptic segmentation.

**Strengths:**

- The paper provides insightful analysis on the importance of the text prompts in the diffusion model under the context of dense prediction tasks.
- The proposed tagging adaptor is easy to implement with standard attention module.

**Weaknesses:**

- The presentation is not clear in many parts
    - Many notations are not explained before being referred in the equation
        - $x$ in Eq. 1, $c$ in Eq 2, $L, D, i$ in Eq. 3, etc.
        - The authors might consider having a separate paragraph explaining the notations
    - It’s unclear what the actual learning objective looks like
    - Overall, it’s hard to parse the details of the architectures and the training recipe.
- Mixed improvements across different tasks
    - Improve in ADE20k and COCO-stuff164k
    - Slightly worse in Cityscapes and COCO-Panoptic

**Questions:**

1. I wonder what leads to the gap between the prompting with the proposed method (56.2 on ADE20k) and ground truth prompting (74.4 on ADE20k). Can the author elaborate more on the intuition of this? To be more specifically, I am curious what kinds of errors cause such a big drop?
2. In sec 3.2, the paper says “Empirical results suggest that the latter approach usually yields enhanced performance.” Do the authors provide the results of freezing the diffusion model parameters somewhere? For fair comparison, ODISE freezes all the diffusion model parameters.
3. From the experimental results, it seems that the propose approach perform well when the number of classes is larger, e.g., ADE20k and COCO-stuff164k. I wonder have the authors tried to train on ADE20k in coarser levels or a subset of ADE20k to see if the improvement diminishes.